# Mathematical and Statistical Evaluation of Reverse Osmosis in the Removal of Manganese as a Way to Achieve Sustainable Operating Parameters

**DOI:** 10.3390/membranes13080724

**Published:** 2023-08-10

**Authors:** Paola Andrea Alvizuri-Tintaya, Esteban Manuel Villena-Martínez, Vanesa G. Lo-Iacono-Ferreira, Juan Ignacio Torregrosa-López, Jaime Lora-García, Paul d’Abzac

**Affiliations:** 1Centro de Investigación en Agua, Energía y Sostenibilidad, Universidad Católica Boliviana San Pablo, La Paz, Bolivia; 2Centro de Investigación de Ingenierías y Ciencias Exactas, Universidad Católica Boliviana San Pablo, Tarija, Bolivia; evillena@ucb.edu.bo; 3Project Management, Innovation and Sustainability Research Center (PRINS), Alcoy Campus, Universitat Politècnica de València, Plaza Ferrándiz y Carbonell, s/n, 03690 Alcoy, Spain; valoia@upv.es; 4Research Institute for Industrial, Radiophysical and Environmental Safety (ISIRYM), Universitat Politècni-ca de València, Plaza Ferrándiz y Carbonell, s/n, 03690 Alcoy, Spain; jitorreg@iqn.upv.es (J.I.T.-L.); jlora@iqn.upv.es (J.L.-G.); 5Centro de Investigación en Ciencias Exactas e Ingenierías, Universidad Católica Boliviana San Pablo, Cochabamba, Bolivia; pdabzac@ucb.edu.bo

**Keywords:** reverse osmosis, safe water, heavy metals, sustainability

## Abstract

Manganese is the Earth’s crust’s third most abundant transition metal. Decades of increased mining activities worldwide have inevitably led to the release of large amounts of this metal into the environment, specifically in water resources. Up to a certain level, manganese acts as an essential micronutrient to maintain health and support the growth and development of microorganisms, plants, and animals, while above a specific limit, manganese can cause toxicity in aquatic and terrestrial ecosystems. There are conventional ways to remove manganese from water, such as chemical precipitation, sorption, and biological methods. However, other treatments have yet to be studied much, such as reverse osmosis (RO), which has demonstrated its effectiveness in the removal of heavy metals and could be a suitable alternative for manganese removal if its energy consumption is reduced. This research presents mathematical and statistical modeling of the behavior of a system in laboratory-scale RO. The principal finding was that it is possible to remove Mn using the RO operated with low pressures without decreasing the sustainable removal efficiency. Reducing the operating costs of RO opens the possibility of implementing RO in different contexts where there are problems with water contamination and economic limitations.

## 1. Introduction

An important part of the environmental degradation suffered by the planet is caused by the contamination of water resources [1]. Metallic ions stand out among other water pollutants, due to their acute toxicity and their carcinogenic nature, denoting a threat to the environment and human health [2,3]. Heavy metals are described as elements with atomic weights between 63.5 and 200.6 and a density greater than 5 g per cubic meter [4]. Some metals, such as copper (Cu), zinc (Zn), manganese (Mn), iron (Fe), and cobalt (Co), play important roles in biochemical processes in the human body. However, excessive exposure to these metal ions can have dangerous impacts. Other heavy metals, such as arsenic (As), cadmium (Cd), lead (Pb), mercury (Hg), and chromium (Cr), are toxic, even at trace levels (parts per billion, ppb), because they are nondegradable and can bioaccumulate in the major systems of the human body [5]. Under normal circumstances, the human body can tolerate small amounts of metal ions without serious problems. However, long-term exposure to these metals can cause high levels of toxin accumulation in the body, leading to failure of body systems and eventually fatality [6,7].

The pollution of water bodies by heavy metals is a global problem [8]. The presence of these metals in water dams and groundwaters is especially dangerous, since this water is the main source of supply for population centers [9,10]. Water with a concentration of heavy metals above the maximum concentration limits (MCLs) stipulated in international standards is not safe for human use or consumption. In this sense, there is a prevailing need to search for treatment technologies that can eliminate heavy metals and guarantee access to safe, quality water for the population.

### 1.1. Manganese and Health Risks

Manganese (Mn) is a natural, almost ubiquitous element found in minerals, rocks, soils, the atmosphere, and different components of the hydrosphere [11]. Up to a certain level, Mn acts as an essential micronutrient to maintain health and support the growth and development of organisms, plants, and animals, while above a certain limit, Mn can cause toxicity in aquatic and terrestrial ecosystems [12]. Based on a compilation of data from large rivers in Africa, Europe, North America, South America, and Asia, dissolved Mn concentrations range from 0.041 ppm to 0.114 ppm [13]. This represents a health risk since the high values found exceed the CML of 0.1 ppm established by international organizations such as the World Health Organization [12] and the United States Environmental Protection Agency [14]. The previous LMC also is adopted in Bolivia [15].

Mn toxicity is known to be linked to adverse effects on the brain, such as neurodegenerative diseases, including Alzheimer’s disease [16]. Excessive accumulation of Mn in the brain can also lead to a neurological syndrome with cognitive, psychiatric, and movement abnormalities, which can also accelerate a specific form of parkinsonism [17,18]. There are even records that Mn in the brain can cause permanent neurological disorders such as manganism, known as “manganic madness”. Cases have been reported in manganese mines in South America, where manganism is considered endemic [19]. Some studies prove the health consequences of children who have ingested unsafe concentrations of Mn. An investigation linked the Mn found in the hair of school-age children with hyperactive behaviors. Exposure to Mn occurred through the consumption of tap water with concentrations outside the CML of the metal [20]. Another study found intellectual disability in children exposed to manganese present in drinking water [21]. The importance of removing Mn from water for consumption is denoted and thus reduces the risks that this metal entails in public health.

### 1.2. Removal of Manganese from Water

Typically, the removal of Mn from water is often driven by potential distribution systems and aesthetic issues. However, Mn removal should be primarily tied to the public health impacts that Mn may have [22]. When selecting a methodology for the removal of Mn, it should be taken into account that the removal of this element is difficult due to its complex chemistry, which makes it a challenge when considering any treatment [23,24]. Additionally, it should be considered that temporal changes in Mn concentrations in streams are complex and may vary depending on the hydrogeological environment [11].

Water treatment technologies to remove metallic ions such as manganese include a wide variety of methods, which can be physical, chemical, biological, or a combination of the above [25]. The classical treatment processes applied for the removal of manganese from water are sorption, redox precipitation, and biochemical processes [11]. Various materials can act as sorption media for Mn; however, it is known that Mn is usually poorly adsorbed [26]. This implies that fixed Mn can be rereleased into water, mainly due to the influence of pH and competing dissolved ions. Therefore, the sorption process alone would not guarantee efficient Mn extraction. The removal of Mn from water through redox precipitation occurs due to the combined effect of sorption and oxidation [11]. Such processes are highly dependent on Eh-pH conditions and dissolved ion content. The newly formed solids can act as scavengers of other dissolved metals and metalloids. However, the high pH requirements for Mn precipitation (pH about 9) may constitute a significant limitation for Mn removal [27]. Finally, the available literature reveals that existing biochemical treatments can efficiently remove manganese, especially when initial concentrations are relatively low [28]. This implies a direct limitation of biochemical techniques when removing high concentrations of Mn from water.

Treatments with high metal ion removal efficiencies are also available, such as the ion exchange processes and membrane filtration [29,30,31,32,33]. However, these treatments are not yet widely applied for the removal of Mn from water. High separation selectivity, in addition to other advantages such as ease of operation and saving time and space, makes membrane processes the most promising technologies for the removal of metallic contaminants from water [3,34,35,36,37]. Figure 1 shows the membrane processes and their differences concerning pore diameter and the main compounds that are retained in each one.

In the last decade, research on the removal of different heavy metals from water by membrane filtration processes has increased significantly: arsenic (III) removal by reverse osmosis (RO) with 98% efficiency [39]; removal of arsenic (V) by nanofiltration (NF) reaching a yield of 90% [40]; lead and cadmium removal, by NF, with 93% and 95% efficiency, respectively [41]; RO application in the removal of lead and cadmium with an efficiency of 98.9% and 99.3%, respectively [42]; reduction in Zn and Cd concentrations by ultrafiltration (UF) with a yield between 92 to 98% [43]; iron removal in UF and RO pilot plant with an efficiency above 94% [44]; removal of chromium (III) by UF reaching a removal yield of 99.5% [45]; and removal of chromium (VI) using RO with an efficiency of 96% [46]. For all these, it is observed that FM processes have high rejection rates for most of the metal cations, highlighting RO as the most promising process. RO can reach efficiencies close to 100%. For this reason, its study and application are gaining importance [47]. In contrast, analyzing RO operating conditions, it is observed that it needs high pressures, which implies a high energy cost for its implementation on an industrial scale.

### 1.3. Reverse Osmosis

RO is the most promising treatment worldwide for the separation of contaminants at the ionic level present in water [48]. However, it has been observed that the high energy costs during its operation have limited its applicability [38]. To establish a mathematical model of the separation process, the concentration-polarization phenomenon was analyzed. The mathematical model allowed us to understand the physical behavior of the system and seek its optimization.

#### 1.3.1. Concentration Polarization

The main reason for the decrease in flux during the initial period of a separation process is the concentration polarization of the solute at the membrane surface [49]. The phenomenon of polarization concentration (CP) that occurs near the membrane significantly affects the mass transfer process. Therefore, the study of mass transfer to the outer membrane focuses on the modeling of the CP phenomenon [50].

The effect of CP in an RO system occurs when the solute tends to accumulate on the membrane surface, preventing the solvent from passing [51]. As the solute separation process progresses in the RO membrane, these are retained within the membrane, and on the internal side of it, these solutes correspond to dissolved salts or contaminating ions that have not been able to pass the permeate. This mass transfer process occurs in the inner boundary layer of the membrane and is called the CP Layer [52].

Figure 2 schematizes the mass transfer model by increasing concentration in the area close to the membrane (inner layer) due to the accumulation of retained solute.

According to the scheme in Figure 2, the system assumes that the flow conditions at distances greater than ẟ inlet solute concentration (C_f_) is constant because the flow is turbulent parallel to the membrane. However, close to the membrane is the boundary layer, where the concentration increases and can reach a maximum at the surface of the C_ẟ1_ membrane. The convective flow of solutes towards the membrane is Jw; if the solute is not completely retained by the membrane, there will be a solute across it, represented as the permeate concentration (Cp) [50,53].

Temperature is a very difficult parameter to control in a natural environment; however, in a plant operating process, it is important to evaluate its effect on the RO operation [54]. For this, the Arrhenius model states that the need for permeability with temperature in RO transport processes must be considered [55]. For this reason, an adjustment was made to the temperature factor that was based on the Arrhenius model to prevent this factor from influencing the response variables.

#### 1.3.2. Spiegler–Kedem Model

According to [56], the Spiegler–Kedem model indicates that the transport of solutes through a membrane can be described with the principles of irreversible thermodynamics (IT), which relate the solvent and solute fluxes with the transport coefficients that in turn are independent of solute concentration [57]. For a system made up of two components, water and solution, IT proposes the following basic equations of the model:(1)Jv=Lp×(Δp−σ×Δπ)
(2)$$Js=B_{s}\times \left(C_{ẟ,1}-C_{p}\right)+\left(1-\sigma \right)\times Jv\times C_{s}=Jv\times C_{p}$$
where:Jv = flux of the solvent.Lp = permeability coefficient of the solvent (water permeability in the membrane).Δp = transmembrane pressure or system operating pressure.σ = reflection coefficient.Δπ = difference in osmotic pressure.Js = solute flow.B_s_ = solute transport coefficient.C_ẟ1_ = solute concentration at the membrane surface (feed side).Cp = concentration of the solute in the permeate.Cs = concentration of the solute inside the membrane.

Several mathematical models have described the mass transfer and hydrodynamic permeability in RO systems [58]. Using a mathematical model that expresses the performance of the RO process is a mechanism that could help to reduce operating costs [59], but it is also necessary to complement the study of the process through statistical analysis, which focuses on the outputs and inputs to know the behavior of the RO and to be able to optimize it [50].

Currently, the optimization of the different water treatments is a key issue in achieving sustainable development, not only from an environmental point of view but also from an economic and ethical point of view [1]. Manganese is a heavy metal identified as a real health and environmental problem in water sources in Bolivia, a developing country that needs to preserve and restore its water sources for a safe supply for its population. The aim of this article is to evaluate through mathematic and statistical methods the RO process operating under critical conditions (low pressures), seeking to reduce its energy cost but without reducing its Mn ion removal efficiency. This will guide RO to a sustainable application that will allow its implementation in contexts with economic limitations.

## 2. Materials and Methods

This section introduces the study area with the problem of Mn contamination in water. The design and construction of the reverse osmosis pilot plant is detailed. The design of the experiments carried out is explained. The preparation of the synthetic water is described. Finally, the mathematical and statistical methods used to evaluate the process of Mn removal by RO are presented.

### 2.1. Study Area

Bolivia has participated in international water forums as a pioneer of the Declaration of Human Rights of Water before the General Council of the United Nations [60,61]. However, despite this, the country still faces several challenges related to water management in practice. The study area of this research is the upper part of the Milluni micro-basin, located at approximately 4600 m above sea level. This micro-basin covers an area of 40 km^2^, is part of the Altiplano basin system, and presents extreme climatic conditions typical of the area [62]. In Milluni, mining activity has been carried out for decades and is currently carried out without all the corresponding controls, which contributes to the degradation of the environment, specifically water [63,64,65]. The lagoons of the upper part of Milluni are important sources of fresh water for La Paz and El Alto, two of the most populated cities in Bolivia [66]. The water treatment plants that treat Milluni’s water do not have specific treatments for the removal of heavy metals; in this sense, the quality of the water is not guaranteed for consumption [9].

The Milluni micro-basin is part of the Katari macro-basin, which is defined as a strategic basin by the Bolivian National Basin Plan [67]. The Katari Basin Management Unit is the body in charge of monitoring the Milluni micro-basin from 2006 to the present. An analysis of Milluni’s historical monitoring data exposes the presence of heavy metals at concentrations outside the permissible limits [9]. Other studies have corroborated the presence of high metal contents in Milluni, including Mn [65,68,69,70]. Figure 3 shows the distribution of the lagoons in the upper part of the micro-basin.

To summarize, the study area of this research is a drinking water supply source in Bolivia that has been affected by ancestral and illegal mining where Mn concentrations exceeding the MCL have been identified, leading to public health risks that must be addressed.

### 2.2. RO Pilot Plant

The RO pilot plant was mounted on an aluminum frame and a stainless-steel table. The system worked in a closed circuit. The synthetic water arranged in the storage tank was pumped to the RO module (membrane). Leaving the module, there were two outlets: the concentrate and the permeate. Both outlets returned the water to the storage tank. Figure 4 presents a scheme of the system.

In the experimental pilot, a commercial membrane model ULP-2540, Keensen, Changsha, China was used [72], with a Wave Cyber, Shanghai, China housing model 2540.300.1. The spirally wound polyamide membrane has an active area of 25 m², which allows a permeate rate of 2.84 mᶟ/d with a concentrate rejection of up to 99.3%, according to manufacturer data. The maximum working pressure that can be applied is 4.14 × 10^6^ Pa, and the minimum concentrate ratio is 8%, according to manufacturer data [72].

Since RO is a process that needs a driving force to carry out the separation process, a multistage centrifugal/electric pump model 2ACM150H, LEO, Zhejiang, China was used. The power of the pump is 2 HP, and it has a maximum operating pressure of up to 5.00 × 10^6^ Pa. In addition, there was a safety valve against possible pressure changes. The pumping system was configured by a frequency regulator that allowed the manipulation of the inlet flow to the system. To protect the membrane module, it was necessary to incorporate a water filter capable of retaining any suspended materials before the pump.

To monitor the inlet and outlet pressures of the system, manometers located before and after the membrane module were used. These allow visualizing the existing pressure loss and controlling the working pressures. For the evaluation of the behavior and operation of the membrane, it was necessary to permanently control the flows of the two outputs of the membrane module (permeate and concentrate). Thus, flow sensors were installed at both outputs. The flow sensors are from the Gems Sensors brand, Plainville, CT, USA; with scales from 0.5 to 5 L/min for the permeate and from 1 to 25 L/min for the concentrate. The sensors were programmed into an Arduino card, which, when connected to a computer, showed the measurement and recording of flow data in real time. In addition, an analog rotameter was installed in the concentrate to have visual control of this output.

For the start-up of the pilot plant, a permeate test was carried out, which provided the reference values for the start of operation in the RO plant. These values served to know if the membrane decreases its permeability over time and to control the effectiveness of the cleaning sessions. Cleaning of the membrane was carried out with citric acid from BIOPACK^®^, El Salvador, Argentina at the end of each experimental test to avoid altering the behavior of the membrane with some type of fouling or incrustation.

### 2.3. Design of Experiments

To determine the number of tests necessary to obtain representative results of the experiment and to be able to evaluate them mathematically and statistically, a design of experiments was necessary. The design was accomplished with the Statgraphics Centurion XVIII program [73] using the “Design of Experiments Assistant” application. To achieve the design of experiments, four steps were followed. First, the output variables were identified with their units in the system. These were: flux analytical medium, “flux” and Rejection Index “R”. As a second step, the input variables were defined. In addition, the variables were of a continuous type, and their values were set for high, medium, and low ranges. The input variables were: pressure “P”, input flow “C”, and solution concentration “C_f_”. As a third step, the type of design to be used was chosen, selecting a factorial fraction with three levels. With this, the number of necessary trials was obtained, which was nine. For each test, the configuration of the input variables was different, enabling evaluation f the process in different scenarios. Finally, the quadratic model was reduced to fit the results of the experiment.

### 2.4. Synthetic Water Preparation

The synthetic water used in the experimentation was prepared by adding a metal solution in distilled water combined with sodium chloride (NaCl). The Mn used was in 1000 μg/mL solutions from Inorganic Ventures^®^, Christiansburg, VA, USA. The NaCl salt used was from BIOPACK^®^ El Salvador, Argentina with a purity > 99%. The concentration levels of the solutions were selected according to the concentrations of Mn and conductivity conditions found in the previous monitoring of natural waters in the study areas. Sampling was carried out in compliance with Bolivian Standard NB-496 [74], which indicates that wide-mouth polyethylene bottles with a capacity of 300 mL should be used. Measurements of Mn concentrations were conducted with an inductively coupled plasma mass spectrometer (ICP-MS), Model 7700×, Agilent Technologies, Santa Clara, CA, USA; Table 1 sets forth the concentrations of Mn and NaCl used in the synthetic waters for the three levels (low, medium, and high) of concentration for the experimentation.

It should be noted that the low level of concentration represents the concentrations identified in the most contaminated point of Milluni. This concentration exceeds by five times what is stipulated for Mn content in international and national standards [12,14,15]. The medium concentration is five times the low concentration, and the high concentration is ten times the low concentration. NaCl was used to replicate the conductivity conditions of the study water.

### 2.5. Mathematical and Statistical Evaluation

The methods used to evaluate the behavior of the RO process are presented below.

#### 2.5.1. Mathematical Method

Jv × C_p_ is expressed as:(3)$$Jv\times C_{p}=Jv\times C-D\frac{dc}{dx}$$
where:C = concentration of the solute in the CP layer.D = diffusion coefficient.

Equation (3) can be transformed into Equation (4) by transposing factors.
(4)$$D\frac{dc}{dx}=Jv\times C-Jv\times C_{p}$$

According to the scheme of Figure 4, the limit conditions are:X = 0 → C = C_ẟ1_
X = ẟ → C = C_f_

Integrate these with the following equation.
(5)$$\frac{C_{ẟ,1}-C_{p}}{C_{f}-C_{p}}=e^{\frac{Jv\times {ẟ}_{cp}}{D}}=e^{\frac{Jv}{k}}$$
where:ẟcp = thickness of the CP layer.k = mass transfer coefficient in the CP.

The principle of conservation of mass is established by Equations (6) and (7).
(6)$$Q_{f}\times C_{f}=Q_{b}\times C_{b}+Q_{p}\times C_{p}$$
(7)$$Q_{f}=Q_{b}+Q_{p}$$
where:Q_f_ = feed flow.C_f_ = feed concentration.Q_b_ = concentrate flow.C_b_ = concentration of the concentrate flow.Q_p_ = permeate flow.C_b_ = concentration of the permeate flow.

The relationship between Q_p_ and Jv is shown in Equation (8).
(8)$$Jv=\frac{Q_{p}}{S}$$
where:S = effective area of the membrane.

The recovery “y”, which represents the water production capacity, is defined as the fraction of the feed flow that passes through the membrane, also called the permeate flow. The greater this flow, the greater the production capacity of the RO system. This value is determined with Equation (9).
(9)$$y=\frac{Q_{p}}{Q_{f}}\times 100$$

The rejection coefficient presented in Equation (10) compares the solute concentration in the inlet flow C_f_ with respect to the solute concentration in the permeate flow C_p_.
(10)$$Ro=\frac{C_{f}-C_{p}}{C_{f}}$$

The reflection coefficient shows the separation capacity of a membrane. For σ = 0, there is no separation, while for σ = 1 there is 100% separation. For this case with a semipermeable membrane, the σ for this study will be close to 1.

Δπ is calculated with Equation (11).
(11)$$\Delta \pi =R\times T\times \left(C_{ẟ,1}-C_{ẟ,2}\right)=R\times T\times (C_{ẟ,1}-C_{p})$$
where:R = constant of the gas law.T = temperature.C_ẟ2_ = concentration of the solute on the surface of the membrane (permeate side).

To calculate the σ factor, it is assumed that the solute concentration in the C_f_ feed is similar to that of the C_ẟ1_ membrane surface [57], so that Equation (12) remains as:(12)$$\Delta \pi =R\times T\times (C_{f}-C_{p})$$

By substituting Equations (5) and (12) in Equation (1), we have Equation (13).
(13)$$Jv=Lp\times \left(\Delta p-\sigma \times \Delta \pi \right)=Lp(T)\times [\Delta p-\sigma \times \Delta \pi \times e^{\frac{Jv}{k}}]$$

The temperature variation during the experimental process generates a variation in the Lp parameter, so it must be adjusted with the Arrhenius model [75], shown in Equation (14):(14)$$A=Ao\times e^{-\frac{\Delta H}{R}\times \left(\frac{1}{T}-\frac{1}{To}\right)}$$

Considering the value of A as the parameter Lp(FaT) adjusted for the adjustment factor for temperature and Ao as the theoretical factor of Lpo. With these considerations, Equation (15) would be as follows:(15)$$Lp\left(FaT\right)=Lpo\times e^{-\Delta H/R\times \left(\frac{1}{T}-\frac{1}{To}\right)}$$
where the adjustment factor is represented according to Equation (16):(16)$$FaT=e^{-\frac{\Delta H}{R}\times \left(\frac{1}{T}-\frac{1}{To}\right)}$$

Combining Equations (13), (15) and (16), Equation (17) is obtained, which will be used for the evaluation of the RO process in this study.
(17)$$Jv=Lpo\times e^{-\frac{\Delta H}{R}\times \left(\frac{1}{T}-\frac{1}{To}\right)}\times [\Delta p-\sigma \times R\times T\left(C_{f}-Cp\right)\times e^{\frac{Jv}{k}}]$$

The −ΔH/R value is calculated from the calibration line of the membrane permeability test carried out with different temperatures. The model predicted a base temperature of 293 K. The model parameters Lp, σ, and k were determined from the experimental data and nonlinear regressions with the calculation and modeling tool Matlab [76].

#### 2.5.2. Statistical Method

Statistical analysis of the results was performed using the Statgraphics Centurion XVIII program [73]. For the evaluation of the results, the Pareto diagram, the ANOVA table, the main effects graph, and the interaction graph were analyzed. Finally, the response surface graph was constructed, which allowed the best combination of work variables to be graphically displayed to identify the optimal work ranges to generate savings during the operation without reducing the RO separation efficiency.

## 3. Results

The input variables of the experiment and results of the experimental part (output variables) are presented in Table 2. The flux data that are presented are calculated with Equation (8) and applying the Arrhenius temperature adjust factor. The rejection index (R) was calculated with Equation (10).

The results obtained were the basis for a mathematical and statistical analysis of the process to maximize the response variables through the combination of critical operating variables. The following sections present the mathematical and statistical evaluation of the results.

### 3.1. Mathematical Evaluation

The behavior of the membrane during the process is presented in Figure 5, where the direct proportionality between flux and pressure is observed. As the concentration increases, the flux decreases, exposing the inverse proportionality between both variables. The results indicate that the microstructure of the membrane would be interacting properly with the solution. On the other hand, it is known that when working with low pressures, the flow behavior with pressure tends to be linear, so no physical compaction of the membrane is observed when solute concentrations are low. For high concentrations, the curve was located below the curves for low and medium concentrations, due to the difficulty in the mass transfer process near the membrane surface.

To understand the physical behavior of the membrane during the experimentation, the effects of the input variables on the responses are analyzed and presented in the following sections.

#### 3.1.1. Effect of Feed Flow (Q_f_) on the flux (Jv)

The feed flow ranges used for the system were 9.95271 × 10^−5^ m^3^/s < Q_f_ < 1.11 × 10^−4^ m^3^/s. The inlet concentrations are shown in Table 1. The temperature variable was considered an adjustment factor for the flux, obtaining that adjustment by the Arrhenius model (Equation (14)).

Figure 6 exposes the behavior of the flux (Jv) concerning the feed flow (Q_f_). It shows that the Jv increases proportionally with the increase in Q_f_. However, it is also observed that with increasing concentration, the Jv decreases. Increasing the Q_f_ with low concentrations increases the velocity, and this increases the mass transfer coefficient and therefore increases the Jv. Conversely, increasing the concentration causes the CP to increase and the Jv to decrease, as described by [50] in a CP process by reverse osmosis of a glucose solution. The foregoing is also represented in Figure 2, where a Qb can be seen, which would be the return flow when the CP phenomenon occurs. Then, it is shown that at low concentrations, the control mechanism is mass transfer, while at high concentrations, the mechanism that governs the process is the concentration polarization.

#### 3.1.2. Effect of Pressure (P) on the Rejection Index (R)

Regarding the efficiency in the removal of the contaminant, Figure 7 shows an adequate behavior of the RO system, since the rejection rates are above 96%. The above in quantitative terms allows establishing that the concentrations of Mn in the permeate are below the parameters established in the national and international regulations for water for human consumption [12,14,15]; therefore, it was demonstrated that the process of RO allows obtaining water with safe concentrations of Mn operating in critical pressure conditions.

Figure 7 shows that the rejection rates are not those expected with the increase in pressure. This refutes what was described by [77], which indicates that with the increase in pressure in a brackish water desalination process by RO, better recovery is expected. Thus, regarding the result of our study, working with low pressures is a possibility that can lead the RO technique toward sustainability without reducing its removal efficiency.

#### 3.1.3. Effect of Solution Concentration (C_f_) on the Flux (Jv) and the Rejection Index (R)

The behavior of the system with respect to the increase in Mn concentrations is analyzed in Figure 8 and Figure 9. Both figures were built with the results obtained for the most critical pressure (5.00 × 10^5^ Pa); however, the same trend is observed for medium and high pressures. The flow rate and temperature were kept fixed.

Figure 8 shows that when Cf increases, Jv decreases. This is explained by the decrease in mass transfer and concentration polarization near the membrane boundary layer.

Although the rejections are higher than 96%, Figure 9 shows a slight decrease in R concerning the increase in C_f_. This can be explained with: as concentration increases, the CP process approaches the limit layer of the membrane, producing a decrease in solute rejection. These results coincide with the work of [56], in which an approach that combines the characterization and modeling of mass transfer in a RO system operated at moderated pressures is presented.

#### 3.1.4. Contrast of Experimental Data vs. Theoretical

For the evaluation of the experimental data vs. the theoretical data, Equation (17) has been used. From this equation. the reflection coefficient σ has been estimated using the curve-fitting application of Matlab [76]. The calculation of nonlinear parameters such as the hydraulic permeability constant Lp and the mass transfer coefficient k, was also performed with Matlab. Table 3 details the coefficients found for the three concentration levels of the solution.

Incorporating the parameters of Table 3 in Equation (17) previously described and explained, the correlation between the time and the values of the theoretical Jv (calculated) and the experimental Jv were determined. The resulting graphs were the tools for evaluating the behavior of the experimental flux concerning the theoretical one and are presented in Figure 10, Figure 11 and Figure 12.

From the previous figures, it can be seen that the maximum difference between the theoretical and experimental values is 6%. The analysis and evaluation of the CP in the membrane has allowed us to validate the efficiency of the system. The temperature correction included in the CP model allows an adequate adjustment of the same, considering that the permeability of the membrane varies according to the increase in temperature of the experimental process. This is observed in the calculated theoretical flow, which shows a clear increasing trend, which is normal in the separation process as the temperature of the solution increases. A congruence between the experimental and theoretical behavior is highlighted, which validates the results of the experimental phase, indicating an adequate operation with the pilot plant.

### 3.2. Statistical Evaluation

The statistical analysis’s most significant result is that operating at low pressures achieves the highest rejection rates. It should be noted that the three pressure levels operated at (5000, 75,000, and 10,000 Pa) were low pressures, in contrast to the pressures that RO processes normally operate. This selection was chosen to observe the technology’s behavior in critical pressure conditions.

As mentioned before, temperature is a factor that could have an impact on the results, as this is a factor that varies with the environment. In this sense, the results presented below were corrected by a temperature correction factor. The response surface graphs are presented below, where the interaction of the three input variables (concentration, pressure, and flow) is observed for each of the response variables (R and flux).

#### 3.2.1. Rejection Index (R)

First of all, it should be noted that Mn rejection rates are above 96%. Considering that low pressure would be the most convenient point to achieve sustainability in the RO technique, the response surface graph was built as a function of this pressure and is presented in Figure 13.

Figure 13 shows that a higher rejection rate would be reached for the low-medium concentration range. This is significant for the study area, since the low level of concentration is five times the concentration of Mn existing in this area, which exceeds national and international standards. Then, a response greater than 96% at the low level would already be solving the simulated contamination problem. Regarding the flow, it can be seen that working with a low, medium, or high flow does not cause a significant direct effect on the response variable R (Figure 13).

#### 3.2.2. Flux (Jv)

Flux represents the flow of water over the active area of the membrane. This response variable was to evaluate the operation process in the RO pilot plant. The flux values obtained in the experimentation were lower than those recommended by the manufacturer. Working with a flux lower than that preset by the manufacturer through the operation with low pressures gave lower electrical consumption without affecting the removal rate (Figure 14).

The evaluation of the behavior of the flux allows us to gain initial notions for the scalability of the RO process. The flux values throughout the experimentation were kept below what is recommended by the manufacturer of the membrane module, which has the following advantages:As already mentioned, operating with low pressures brings energy savings that would not decrease the removal efficiency of the membrane modules.The longer useful life of the equipment, since it will not work at full capacity, leading to an increased lifetime of the membranes.The periods between equipment maintenance will be longer, which implies not having as many problems with membrane fouling, which also means fewer chemical cleanups.

These point to possibilities of reducing operating costs in an RO process if working with flux values lower than those stipulated by the supplier.

## 4. Discussion

Taking into account that surface water resources are not infinite, maintaining water quality is a necessity for the survival of living beings. Due to its complex chemistry, Mn presents a challenge for classical water treatment techniques, which has led to the study of other alternatives such as RO. RO stands out for its high efficiency, without the need for chemicals, without the production of solid waste, and simple and compact automation [38,40,42,78]. Despite the above, much of the recent research on Mn treatment by RO has been conducted at the laboratory level and remains limited [78]. In this sense, experimental work with RO pilot plants becomes fundamental to simulate operating conditions close to reality and thus obtain representative information on the behavior of the process to achieve adequate and sustainable scaling.

The mathematical evaluation of the process exposed that the recovery of water (flux) increases with the increase of the pressure and temperature of feeding, but it decreases with the increase in concentration of the solution, in agreement with the work of [77]. The permeability in the transport of solutes followed an Arrhenius-type relationship, as indicated by [55], which allowed analyzing the effect of temperature on the functioning of the RO. Regarding the rejection index (R) values, these decrease as the level of concentration increases, which is normal behavior according to the work of [56]. Despite working with low pressures, it was confirmed that at low concentrations, the control mechanism is mass transfer, while at high concentrations, the mechanism that governs the process is concentration polarization, corroborating the study of [50]. The comparison between the experimental and theoretical values by a mathematical model helps us to physically understand the process and identify the optimal operating conditions, in congruence with the research carried out by [50].

According to [79], statistical models provide a wide range of information that allows a statistical prediction of the operation of the plant and improved efficiency in its development. These inputs are useful in RO plants’ planning, operating, and monitoring processes. This study produced a statistical evaluation to identify the best configuration of the operating variables to maximize responses in an RO pilot plant, seeking to reduce operating costs. This is an important step before scaling up to achieve sustainable systems, which other studies highlight [80].

## 5. Conclusions

This investigation concludes that the RO process operated at low pressures is efficient in removing Mn from polluted water. The mathematical and statistical evaluations of the RO process allowed us to understand the physical behavior of the membrane and identify the operating ranges without decreasing the sustainable removal efficiency. The rejection rates with the pilot plant for the most critical pressure conditions are above 96%.

Operating with low pressure contributes to reducing the energy cost during operation and also has an impact on reducing flux. Working with flux values below that stipulated by the manufacturer implies less chemical cleaning of the membranes and a smaller number of membrane replacements over time, favoring operating costs in the same way. All of the above has a significant impact on a massive implementation of RO and is significant in contexts with economic limitations, but with imminent risks to public health and ecosystems, as in the case of water contamination by Mn in Milluni, Bolivia.

RO is a promising water treatment technique because it produces safe water with high quality and reduces the volume of contaminants. Further research can use the results of this research to implement large-scale RO plants with sustainable operation. In addition, in the RO process, the contaminants are concentrated in one outlet; the recovery of Mn could be carried out more easily through other processes. That would favor the prevailing need to recover Mn due to its negative implications for public health and ecosystems.

## Figures and Tables

**Figure 1 membranes-13-00724-f001:**
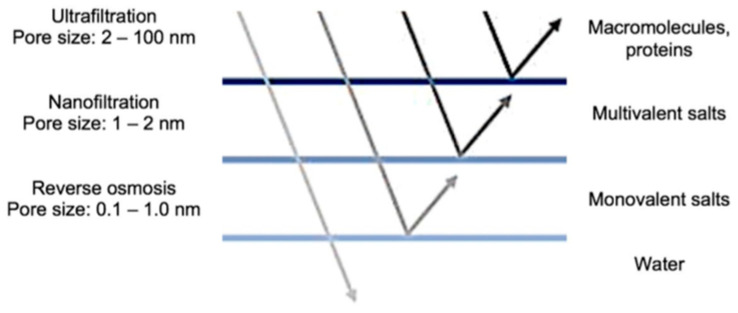
Membrane filtration processes. Source: Redrawn from [38].

**Figure 2 membranes-13-00724-f002:**
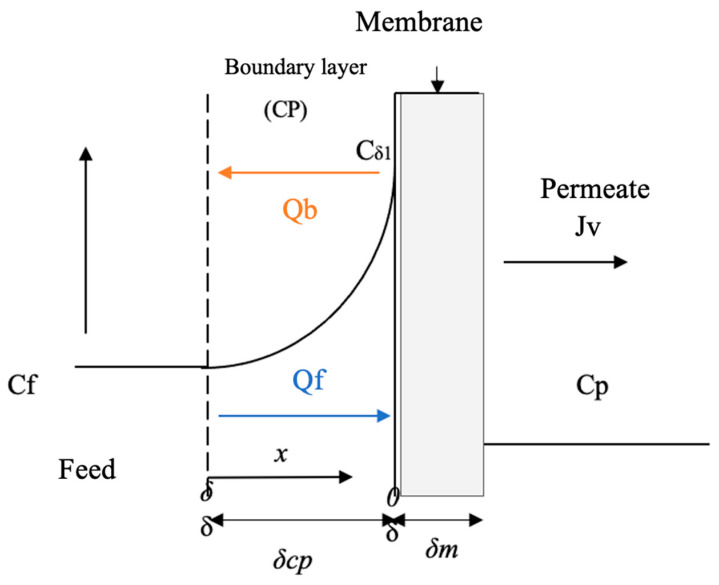
Concentration-polarization scheme in RO. Source: Own elaboration based on [49].

**Figure 3 membranes-13-00724-f003:**
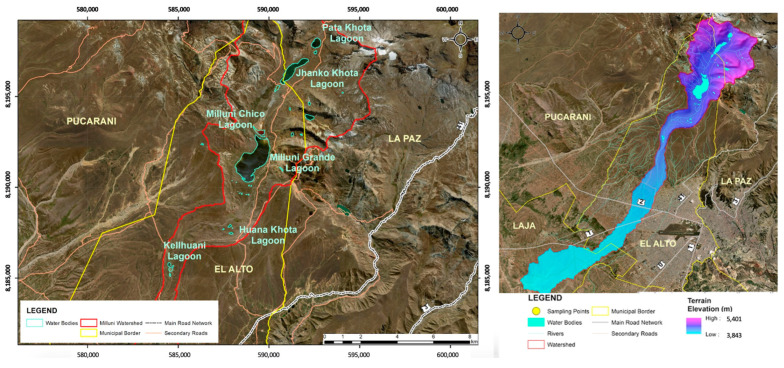
Location Milluni and its lagoons. Source: Redrawn from [68,69].

**Figure 4 membranes-13-00724-f004:**
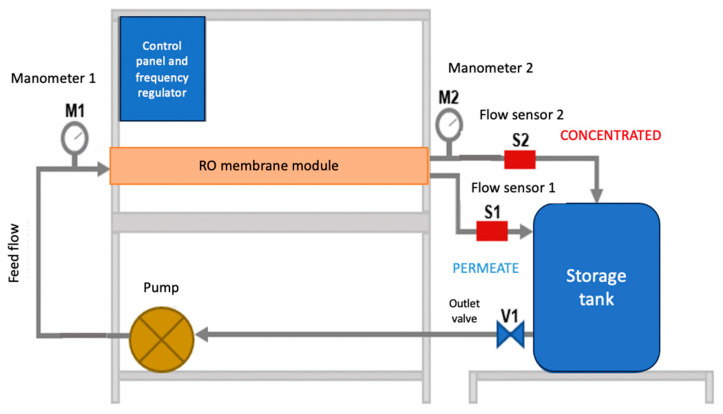
RO pilot plant scheme. Source: Redrawn from [71].

**Figure 5 membranes-13-00724-f005:**
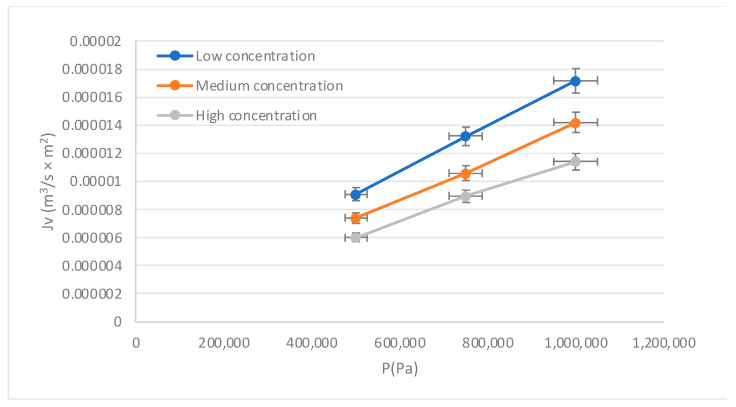
Behavior of flux (Jv) on the pressure (P). Source: Own elaboration, 2023.

**Figure 6 membranes-13-00724-f006:**
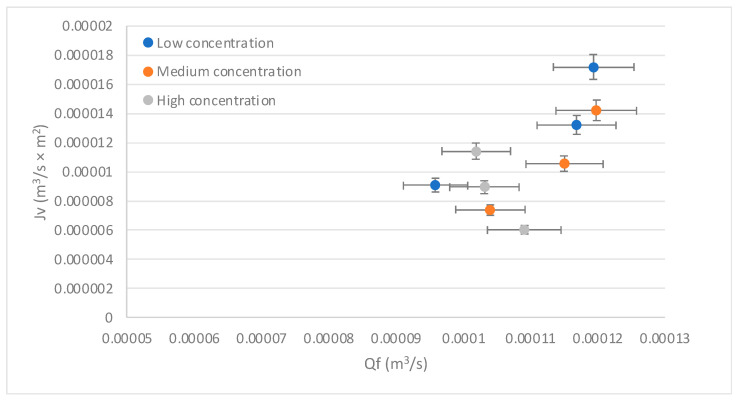
Behavior of the flux (Jv) on the feed flow (Q_f_). Source: Own elaboration, 2023.

**Figure 7 membranes-13-00724-f007:**
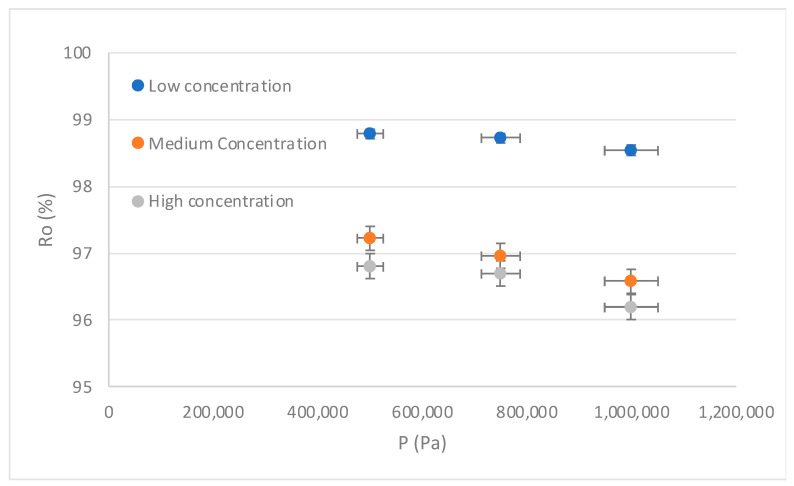
Influence of pressure (P) on the rejection index (R). Source: Own elaboration, 2023.

**Figure 8 membranes-13-00724-f008:**
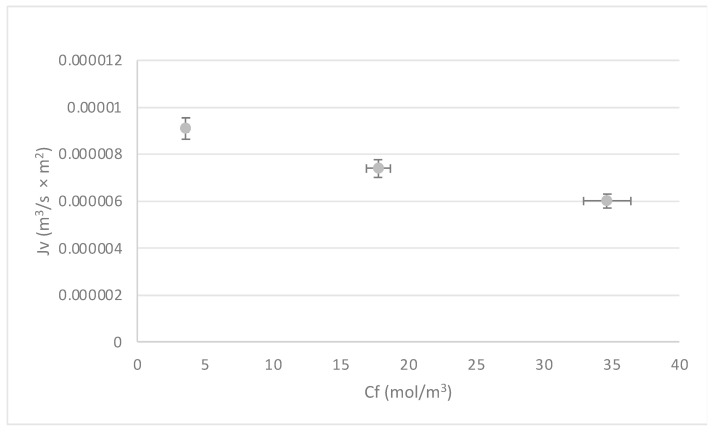
Influence of the solution concentration (C_f_) on the behavior of the flux (Jv). Source: Own elaboration, 2023.

**Figure 9 membranes-13-00724-f009:**
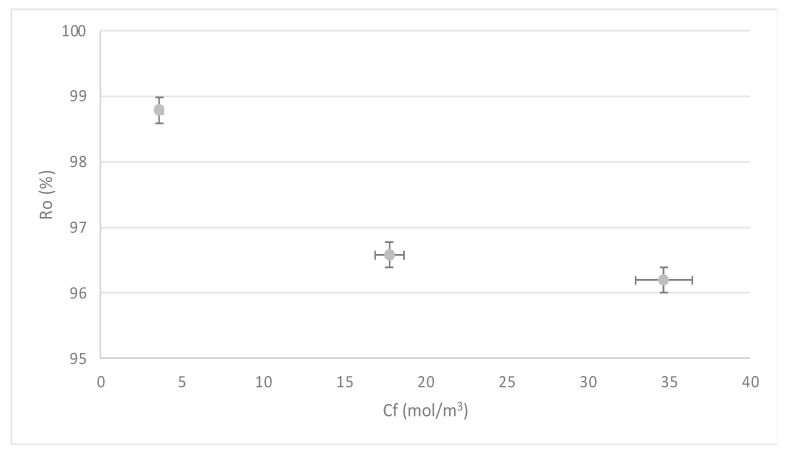
Influence of the solution concentration (C_f_) on the behavior of the rejection index (R). Source: Own elaboration, 2023.

**Figure 10 membranes-13-00724-f010:**
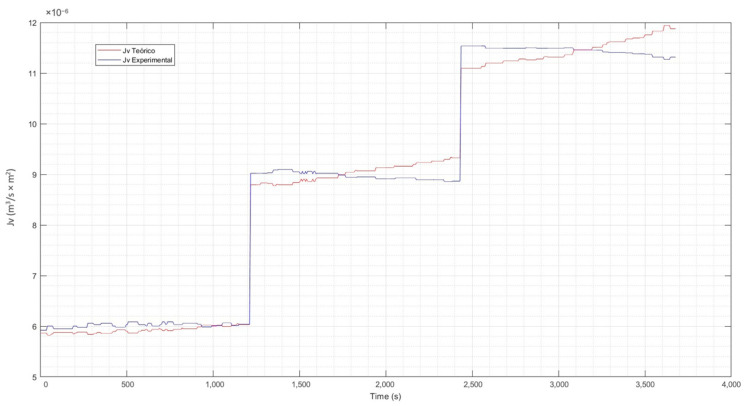
Theoretical flux vs. experimental flux at low concentration. Source: Own elaboration, 2023.

**Figure 11 membranes-13-00724-f011:**
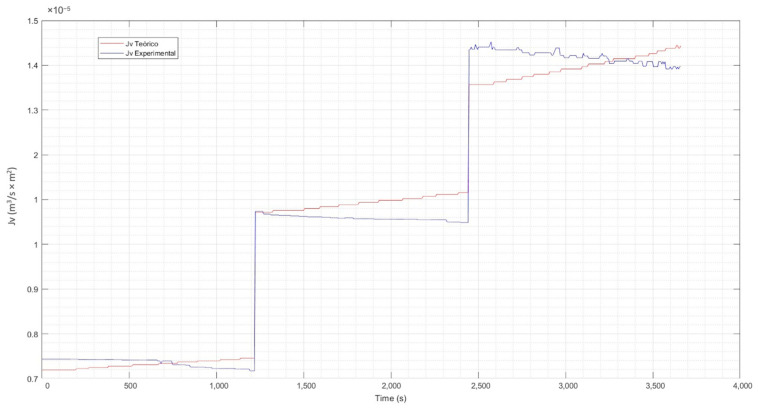
Theoretical flux vs. experimental flux at medium concentration. Source: Own elaboration, 2023.

**Figure 12 membranes-13-00724-f012:**
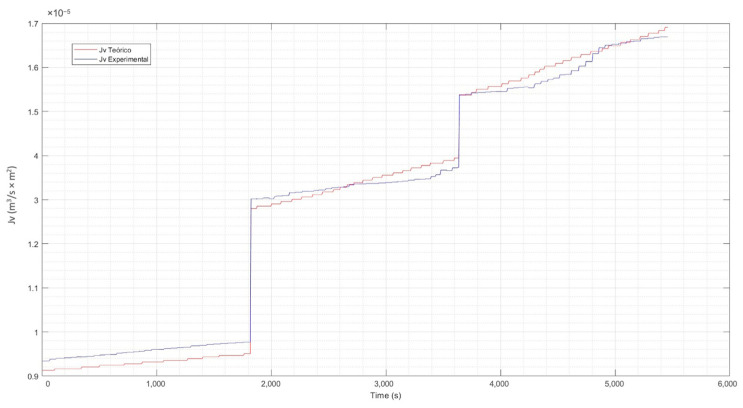
Theoretical flux vs. experimental flux at high concentration. Source: Own elaboration, 2023.

**Figure 13 membranes-13-00724-f013:**
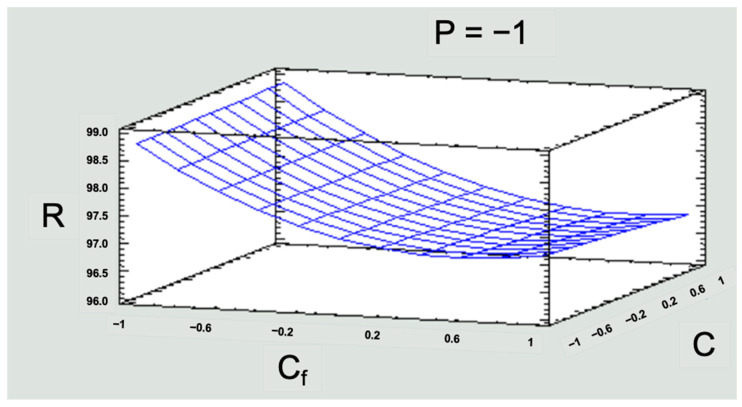
Response surface graph for R in the low-pressure operating scenario. Note 1: R = rejection index; P = pressure; Cf = solution concentration; C = feed flow. Note 2: for the three variables (P, Cf, C), −1 is the low level, 0 is the medium level, and 1 is the high level. Source: Own elaboration, 2023.

**Figure 14 membranes-13-00724-f014:**
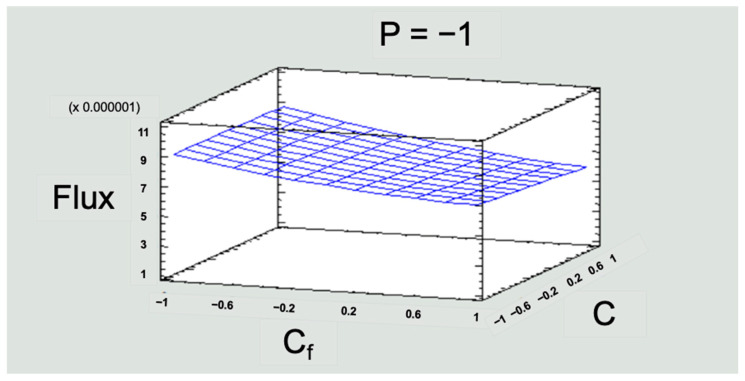
Response surface graph for flux in the low-pressure operating scenario. Note 1: P = pressure; Cf = solution concentration; C = feed flow. Note 2: for the three variables (P, Cf, C), −1 is the low level, 0 is the medium level, and 1 is the high level. Source: Own elaboration, 2023.

**Table 1 membranes-13-00724-t001:** Synthetic water concentrations.

Solution Concentration	Mn (mol/m^3^)	NaCl (mol/m^3^)	Solution Concentration Cf (mol/m^3^)
Low	1.08 × 10^−2^	3.59 × 10^0^	3.60 × 10^0^
Medium	5.38 × 10^−2^	1.77 × 10^1^	1.78 × 10^1^
High	1.04 × 10^−1^	3.46 × 10^1^	3.47 × 10^1^

Source: Own elaboration, 2023.

**Table 2 membranes-13-00724-t002:** Results of experimentation. Source: Own elaboration, 2023.

No	Input Variables			Output Variables	
	Solution Concentration Cf (mol/m^3^)	Pressure P (Pa)	Income Flow C (m^3^/s)	Flux Jv (m^3^/s × m^2^)	Global Rejection Rate R (%)
1	3.60 × 10^0^	5.00 × 10^5^	9.57 × 10^5^	9.09 × 10^−6^	98.79
2	3.60 × 10^0^	7.50 × 10^5^	1.17 × 10^−4^	1.32 × 10^−5^	98.72
3	3.60 × 10^0^	1.00 × 10^6^	1.13 × 10^−4^	1.72 × 10^−5^	98.54
4	1.78 × 10^1^	5.00 × 10^5^	1.04 × 10^−4^	7.39 × 10^−6^	97.22
5	1.78 × 10^1^	7.50 × 10^5^	9.95 × 10^−5^	1.06 × 10^−5^	96.96
6	1.78 × 10^1^	1.00 × 10^6^	1.20 × 10^−4^	1.42 × 10^−5^	96.58
7	3.47 × 10^1^	5.00 × 10^5^	1.09 × 10^−4^	6.01 × 10^−6^	96.80
8	3.47 × 10^1^	7.50 × 10^5^	1.03 × 10^−4^	8.95 × 10^−6^	96.69
9	3.47 × 10^1^	1.00 × 10^6^	1.02 × 10^−4^	1.14 × 10^−5^	96.19

**Table 3 membranes-13-00724-t003:** Model parameters.

Solution Concentration	σ	Lp (m^3^/s × m^2^ Pa)	K (m/s)
Low	0.9850	1.5822 × 10^−11^	5.6 × 10^−6^
Medium	0.9908	1.4160 × 10^−11^	5.83 × 10^−6^
High	0.9893	1.9043 × 10^−11^	7.49 × 10^−6^

Source: Own elaboration, 2023.

## Data Availability

Not applicable.

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
