# Peer review of "Mathematical and Statistical Evaluation of Reverse Osmosis in the Removal of Manganese as a Way to Achieve Sustainable Operating Parameters"

_membranes, 2023, doi:10.3390/membranes13080724_

Round 1

Reviewer 1 Report

-       The current work is related to the Mathematical and statistical evaluation of Reverse Osmosis, but no explanation is given in the introduction about the published models. Additional explanations have been proposed in the introduction. It is necessary to revise the introduction completely.

-       Fig. 5 to Fig. 9 should be improved. These Figs are not of the required quality.

-       Give an uncertainty analysis. Error bars should be added to the Figures.

-       The effect of concentration polarization (CP) and fouling should be considered. These phenomena are so important in the RO process but there is no explanation or any attention to them. The performance of the membrane decreases with time due to fouling and CP.

-       Explanation of Fig. 6 is not enough. It should be described in more detail.

it recommended to revise the manuscript. 

Author Response

Dear reviewer, 

Thanks for taking the time to review our paper. We appreciate your comments. We implemented all your recommendations as we think that they significantly improve our work. For an easier review, please, find below our answers to each of your comments and indications on where to find the changes in the new manuscript.  

We hope you find the improvements adequate.

Kind regards,  

The authors

………………………………………………...………………………………………………

  1. The current work is related to the Mathematical and statistical evaluation of Reverse Osmosis, but no explanation is given in the introduction about the published models. Additional explanations have been proposed in the introduction. It is necessary to revise the introduction completely.

The observation was attended.

  1. Fig. 5 to Fig. 9 should be improved. These Figs are not of the required quality.

The figures' quality has been improved.

  1. Give an uncertainty analysis. Error bars should be added to the Figures.

Error bars added.

  1. The effect of concentration polarization (CP) and fouling should be considered. These phenomena are so important in the RO process but there is no explanation or any attention to them. The performance of the membrane decreases with time due to fouling and CP.

The effects of concentration-polarization (CP) and fouling were taken into account in the present investigation. The CP effect is manifested when we compare the flux of distilled water against the Flux of the solution of salts. Within the mathematical model, it is collected by the influence of the concentration (Cf) and the flow of water that affects the mass transfer coefficient of the water through the polarization layer.

Throughout all the experiments it has been argued that when working with low concentration of metals and with high flow rates, the CP effect is significantly reduced, while when the concentration rises and the flow rate decreases, the CP effect is more noticeable.

Regarding fouling in the results obtained, its presence can be considered negligible because we worked with model solutions, and the operation time was reduced. It is also important to point out that when working with a membrane module with an active area of ​​2.5 m^2, rapid system stability was achieved in all cases, which helps to better interpret the results.

This is consistent with the fact that when working with large active areas, the CP phenomenon takes longer to occur. In addition, it should also be taken into account that a synthetic solution was used, which did not favor the fouling of the membrane module; this was confirmed by not presenting a drop in Flux after each experimental run. In conclusion, the study of membrane fouling may be more significant when working with real water and for longer experimental periods. This was included in discussions of the results to clarify the observation made.

  1. Explanation of Fig. 6 is not enough. It should be described in more detail.

The explanation of Figure 6 has been completed.

Reviewer 2 Report

In this article, the operating parameters of reverse osmosis system in manganese removal were investigated by mathematical and statistical evaluation. This work effectively helps us understand the physical behavior of reverse osmosis membrane in removing manganese, which is of great significance for determining reasonable operating parameters and reducing operating costs. This article is recommended to be published on membrane, but before that, some key issues need to be explained more specified.

1.              Compared with sorption, redox precipitation, and biochemical processeshow much operating costs can be reduced by using reverse osmosis membranes for the manganese removal?

2.              In Figure 6, why did Jv show such a significant decrease when Qf increased from about 0.000113 to 0.000117 at low concentration?

3.              In the section 3.1.3, the effect of solution concentration (Cf) on the Flux (Jv) and the rejection index (R) were investigated. Whether other operating parameters like Qf, temperature and pressure were kept unchanged during the exploration. Please provide additional explanations in the manuscript.

Some statements are suggested to be well revised. 

Author Response

Dear reviewer, 

Thanks for taking the time to review our paper. We appreciate your comments. We implemented all your recommendations as we think that they significantly improve our work. For an easier review, please, find below our answers to each of your comments and indications on where to find the changes in the new manuscript.  

We hope you find the improvements adequate.

Kind regards,  

The authors

………………………………………………...………………………………………………

1. Compared with sorption, redox precipitation, and biochemical processes, how much operating costs can be reduced by using reverse osmosis membranes for the manganese removal?

It is difficult to make an economic comparison between technologies if they are not studied within the same context. Reverse osmosis is a technology with great efficiency in separating contaminants but, on the other hand, it is expensive both in its implementation and during its operation due to its energy intensity. This research was developed to reduce operating costs so that RO can be competitive (operating at low pressures) compared to other traditional treatments such as sorption, redox precipitation, and biochemical. The application of RO continues to be a challenge in contexts with economic limitations but with serious problems of contamination by heavy metals. In this sense, reducing RO operating costs contributes to providing safe water to populations.

2. In Figure 6, why did Jv show such a significant decrease when Qf increased from about 0.000113 to 0.000117 at low concentration?

Thank you very much for your observation, thanks to your observation it was possible to correct graph 6 since an error was made when transferring the data obtained to the graph, for this reason, the aforementioned decrease was observed. The graph has been replaced and it can now be clearly seen that increasing Qf with low concentrations increases velocity, and this increases the mass transfer coefficient and therefore increases Jv.

3. In the section 3.1.3, the effect of solution concentration (Cf) on the Flux (Jv) and the rejection index (R) were investigated. Whether other operating parameters like Qf, temperature and pressure were kept unchanged during the exploration. Please provide additional explanations in the manuscript.

The aforementioned parameters were considered, but the emphasis was placed on analyzing the separation efficiency when operating with low pressure since the pressure parameter has a direct incidence on the energy consumed during RO operation, affecting the operating costs of technology. The script was implemented at the beginning of section 3.1.3. with the corresponding information.

Round 2

Reviewer 1 Report

Accept